# Molecular characterization of a Novel NAD⁺-dependent farnesol dehydrogenase *SoFLDH* gene involved in sesquiterpenoid synthases from *Salvia officinalis*

**Mohammed Ali**[1☯], **Elsayed Nishawy**[1☯], **Walaa A. Ramadan**[2], **Mohamed Ewas**[1], **Mokhtar Said Rizk**[1], **Ahmed G. M. Sief-Eldein**[1], **Mohamed Abd S. El-Zayat**[1], **Ahmed H. M. Hassan**[1], **Mingquan Guo**[3,4], **Guang-Wan Hu**[3,4], **Shengwei Wang**[5], **Fatma A. Ahmed**[6], **Mohamed Hamdy Amar**[1*], **Qing-Feng Wang**[4,5]

**1** Department of Genetic Resources, Desert Research Center, Cairo, Egypt, **2** Genetics and Cytology Department, Biotechnology Research institute, National Research Centre, Giza, Egypt, **3** Key Laboratory of Plant Germplasm Enhancement and Specialty Agriculture, Wuhan Botanical Garden, Chinese Academy of Sciences, Wuhan, China, **4** Sino-Africa Joint Research Center, Chinese Academy of Sciences, Wuhan, China, **5** Hubei Minzu University, Enshi, China, **6** Department of Medicinal and Aromatic Plants, Desert Research Center, Cairo, Egypt

☯ These authors contributed equally to this work.

* mohamed.amar@wbgcas.cn

**Data Availability Statement:** All relevant data are within the manuscript and its Supporting information files.

## Abstract

*Salvia officinalis* is one of the most important medicinal and aromatic plants in terms of nutritional and medicinal value because it contains a variety of vital active ingredients. Terpenoid compounds, particularly monoterpenes (C10) and sesquiterpenes, are the most important and abundant among these active substances (C15). Terpenes play a variety of roles and have beneficial biological properties in plants. With these considerations, the current study sought to clone theNAD+-dependent farnesol dehydrogenase (SoFLDH, EC: 1.1.1.354) gene from *S. officinalis*. Functional analysis revealed that, *SoFLDH* has an open reading frame of 2,580 base pairs that encodes 860 amino acids.*SoFLDH* has two conserved domains and four types of highly conserved motifs: YxxxK, RXR, RR (X8) W, TGxxGhaG. However, *SoFLDH* was cloned from *Salvia officinalis* leaves and functionally overexpressed in *Arabidopsis thaliana* to investigate its role in sesquiterpenoid synthases. In comparison to the transgenic plants, the wild-type plants showed a slight delay in growth and flowering formation. To this end, a gas chromatography-mass spectrometry analysis revealed that *SoFLDH* transgenic plants were responsible for numerous forms of terpene synthesis, particularly sesquiterpene. These results provide a base for further investigation on *SoFLDH* gene role and elucidating the regulatory mechanisms for sesquiterpene synthesis in *S. officinalis*. And our study paves the way for the future metabolic engineering of the biosynthesis of useful terpene compounds in *S. offcinalis*.

**Funding:** This study was funding by the National Natural Science Foundation of China and with Academy of Scientific Research and Technology (ASRT) under the collaboration project between Wuhan Botanical Garden, Chinese Academy of Sciences, Wuhan, 430074, China, and Desert Research Center (DRC), Cairo, Egypt.

**Competing interests:** The authors have declared that no competing interests exist.

**Abbreviations:** EOs, Essential oils; OE, Overexpression; Semi-RT-PCR, Semiquantitative RT-PCR; *SoFLDH*, *S. officinalis* NAD+-dependent farnesol dehydrogenase; TPS, Terpene synthase.

## Background

The genus Salvia (Lamiaceae) includes more than (>1,000) species of woody aromatic shrubs, among which e.g., *S. epidermindis*, *S. japonica*, *S. fruticosa*, *S. tuxtlensis*, *S. miltiorrhiza*, *S. aureus*, *S. przewalskii*, *S. hydrangea*, *S. isensis*, *S. tomentosa*, *S. santolinifolia*, *S. lavandulifolia*, *S. chloroleuca*, *S. glabrescens*, *S. nipponica*, *S. macrochlamys*, *S. allagospadonopsis* and *S. recognita* are economically important and cultivated worldwide for its medical properties and their output of essential oils (EOs). Most of *Salvia* plant species are commonly spread universally in three areas but largely grown in South and Central America (~500 species), East and West Asia (~100 and ~200 species) respectively [1–4] Lately, *Salvia* species EOs have a valued source for aromatic and medical research for discovering and identifying the ingredient-compounds [3–5]. Essential oils of salvia species have antimicrobial, anticancer, anti-inflammatory, antioxidant, choleretic and antimutagenic properties [6–8].

It is well-known that Terpenoid form is the major cluster of natural compounds and a set of secondary metabolites, that have been identified from plant kingdom and other organisms with more than (>40,000) different structures [9]. Terpenoid derives its structural from the isopentenyl diphosphate (IPP), which involves five carbon atoms (C5) [10, 11]. However, the origin name of these different structures arises from the terebinth tree (*Pistacia terebinthus*), all of these identified as different structures terpene. The structure of the terpene unit was illustrated by Wallach and Altered by Ruzicka [12–15]. Some terpenes are related to the plant primary metabolism such as the carotenoid pigments, phytol side chain of chlorophyll, gibberellin plant hormones, and phytosterols of cellular membranes [16, 17], which are important for development and flower blooming of the plants. However, wide ranges of terpenes have been identified as categories of secondary metabolites with essential properties in the adaptation of plants to the stresses. Nonvolatile and volatile terpenes have an important role in the predators of herbivores and defense against photo-oxidative stress, haul of both pollinators and the direct defense against insects and microbes [18]. At present, several studies are focused to understand in-depth the mechanisms of terpene and its functions [3, 4].

It is prominent that salvia species involved high proportion of the essential oil; this fragrant oil mainly contains monoterpenes and sesquiterpenes. The composition of the monoterpenes and sesquiterpenes are differ depends on the plant species and cultivars, in addition to the type of tissues [3, 4, 19–24]. Biosynthesis gene for the sesquiterpene has remained elusive until recently.

The main sesquiterpene in the Egyptian cultivar of *S. officinalis* are Isocaryophyllene, α-caryophyllene, Caryophyllene oxide and (−)-Germacrene D, which are encoded by three unigenes families [3]. So far, their biological or physiological functions have been widely unclear. This makes the enzymes that stimulate the formation both interesting and functionally difficult to differentiate.

Through this research, we focus to clone and functionally expressed the *S. officinalis* NAD+-dependent farnesol dehydrogenase (*SoFLDH*, EC: 1.1.1.354) gene in *Arabidopsis thaliana*. The recombinant *SoFLDH* catalyzed the conversion of Farnesyl pyrophosphate (FPP) to the sole produce various types of terpene especially sesquiterpene. SoFLDH protein displayed a distinctive amino acid sequence, with highly preserved motifs, including the YxxxK, RXR, RR (X 8) W and TGxxGhaG motifs. Finally, this study reveals to use the protein modeling database to investigate the performance of the 3D structure protein and its function predict.

## Materials and methods

### Plant materials and tissue collection

*Salvia officinalis* seeds were kindly provided by the staff member of Egyptian Desert Gene Bank (EDGB) of Desert Research Center (DRC), Egypt. *S. officinalis* seeds had been growing in our growth chamber at National Research Centre (Cairo, Egypt), at temperature of 22ºC day/20ºC night with humidity of 50–70%, and photoperiod at 16 h day/8 h night, with a light density of 100–150 µmoles m 2‾s 1‾ using fluorescent bulbs. For gene amplification, young leaves were picked up and instantly cast in liquid nitrogen and stowed at −80 ˚C until RNA extraction.

### *In silico* analysis of *SoFLDH*

*SoFLDH* nucleotide sequence was carefully selected from our previous RNA-Seq data [3]. The physiochemical assets of the *SoFLDH* were gritty using PROTPARAM website (http://web. expasy.org/protparam/). Putative tissue expression profile and cell subcellular localizations for *SoFLDH* gene was built using ePlant and cell eFP (http://bar.utoronto.ca/eplant/& http://bar. utoronto.ca/cell_efp/cgi-bin/cell_efp.cgi) based on *Arabidopsis* gene expression and protein localization at different tissues and cell organs. The open reading frames (ORF) for *SoFLDH* was analyzed for the presence of possible transit peptide using bioinformatics tools, iPSORT Prediction (http://ipsort.hgc.jp/). Sequence analysis of *SoFLDH* was executed using NCBI BLAST in contradiction of the protein database (https://blast.ncbi.nlm.nih.gov/Blast.cgi). Clustal Omega software was used with the default parameters (https://www.ebi.ac.uk/Tools/ msa/clustalo/) for multiple sequence alignment. Three-dimensional (3D) structure for SoFLDH protein was built using SWISS-MODEL (https://swissmodel.expasy.org) website based on the closest homologous structures. A maximum likelihood tree was constructed for *SoFLDH* gene using MEGA 6.6 program with default parameters. To assess the phylogeny of the SoFLDH protein sequence in relation to other orthologous terpene alcohol dehydrogenases and benzyl alcohol dehydrogenases genes, the protein sequences of characterized terpene alcohol dehydrogenases and benzyl alcohol dehydrogenases were regained from the National Center for Biotechnology Information (NCBI) database, and we looked for the other reported full-length sequences of terpene alcohol dehydrogenases and benzyl alcohol dehydrogenases from *Salvia splendens* (farnesol dehydrogenase; TEY48599.1), *Persicaria minor* (nerol dehydrogenase; AFQ59973.1), *Salvia splendens* (hypothetical protein Saspl_009804; TEY79171.1), *Carpoglyphus lactis* (geraniol dehydrogenase; BAG32342.1), *Castellaniella defragrans* (geraniol dehydrogenase; CCF55024.1), *Arabidopsis thaliana* (Rossmann-fold NAD(P)-binding domain-containing protein; AEE86213.1), *Ocimum basilicum* (geraniol dehydrogenase; AAX83107.1), *Aedes aegypti* (NADP+-dependent farnesol dehydrogenase; ADB03640.1), *Fragaria x ananassa* (cinnamyl alcohol dehydrogenase; AAD10327.1), *Artemisia annua* (cinnamyl alcohol dehydrogenase; ACB54931.1), *Mentha x piperita* ((-)-isopiperitenol dehydrogenase; AAU20370.1), *Lavandula x intermedia* (borneol dehydrogenase; AFV30207.1), *Pseudomonas putida* (p-cumic alcohol dehydrogenase; AAB62297.1), *Nicotiana tabacum* (allyl alcohol dehydrogenase; BAA89423.1), *Arabidopsis* thaliana (allyl alcohol dehydrogenase; AAG50689.1), *Pseudomonas putida* (aryl alcohol dehydrogenase; P39849.1), *Picea abies* (cinnamyl alcohol dehydrogenase; CAA0597.1), Streptomyces sp. NL15-2K (coniferyl alcohol dehydrogenase; BAN09098.1), and *Pseudomonas putida* (benzyl alcohol dehydrogenase; AAC32671.1) [25].

## RNA extraction and cDNA library preparation

Young leaves of *S. officinalis* were used to extract the total RNA using TransZol Reagent (Focus Bioscience, Australia) according to the manufacturer's instructions and cured with DNase I (Takara). RNA quality was performed on 1.4% Agarose gel, and the clarity was analyzed using a Nanodrop ND1000 (NanoDrop technologies, Wilmington, DE, USA). RNA pools were primed for cDNA libraries using mixing equal volumes from the three RNAs replications in one tube. Two micrograms of total RNA (800 ng approximately) per sample was used for the synthesis of total cDNA with TransScript® First-Strand cDNA Synthesis Super-Mix (TransGen Biotech, Beijing, China) according to the manufacturer's instructions. Afterwards, PCR was performed for cDNA synthesis at 42˚C for 15 min followed by 85˚C for 5 min [3, 4].

## QRT-PCR, semiquantitative RT-PCR analysis and Western Blot (WB)

Quantitative RT-PCR was performed by an IQTM5 Multicolor Real-Time PCR Detection System (Bio-Rad, USA) as described previously Ali et al., and Hussain et al., [3, 4, 26, 27] with SYBR Green Master (ROX) (Newbio Industry, China) following the manufacturer's instructions at a total reaction volume of 20 μl. A gene-specific primer for *SoACTIN* forward 5′– GGCAGTTCTCTCCCTCTAT–3′ and reverse 5′– GAGGTGGTCGGTGAGAT–3′ was used as a reference gene with 157 bp, and *SoFLDH* forward 5′– TTCCTGATCCCTCCAGATT–3′ and reverse 5′– CAATGTAGCCATCCGTTGA–3′ with 153 bp length. Moreover, semiquantitative real-time PCR was achieved on a Biometra PCR (Biometra T Gradient Thermo block PCR Thermocycler, American Laboratory Trading, San Diego, CA) system with a total reaction volume of 25 μl. A gene-specific primer for *At-B-actin* forward 5′–GGCTGAGGCTGATGA TATTC–3′ and reverse 5′–CCTTCTGGTTCATCCCAAC –3′ was used as a reference gene with 155 bp and the same forward and reverse primers for *SoFLDH*, all the primers were designed using the primer designing tools of IDTdna (http://www.idtdna.com/scitools/ Applications/RealTimePCR/). The semiquantitative RT-PCR conditions were as follows: pre-denaturation step at 95˚C for 4 min, 35 cycles of amplification (95˚C for 30 s, 58˚C for 30 s and 72˚C for 1 min), and a final extension step at 72˚C for 10 min. The PCR products were resolved on 1.3% agarose gel, and the expression levels of *At-B-actin* and *SoFLDH* genes were detected. On the other hand, for confirmed the transformation stability of *SoFLDH* protein was isolated and detected from various lines of transgenic and wild-type *A. thaliana* using Western blotting (WB), fresh leaves were homogenized with pestles in prechilled mortars on ice in 1 ml of cold homogenization buffer (100 mM Tris, pH 7.5, 10% sucrose, 5 mM sodium EDTA, and 5 mM sodium EGTA) for each one gram of tissue as described by Ma et al., [28].

## Full-length terpene synthase cDNA clone and vector

Full-length cDNAs sequence for *SoFLDH* was obtained based on RNA-Seq sequence information from our transcriptome sequencing of *S. officinalis* plant leaves [3], and *SoFLDH* was amplified from cDNA of young leaf using short and long gene-specific gene primers based on the Gateway pDONR221 vector manual system. The initial PCR amplification was performed by short primers, of *SoFLDH* forward 5′– ATGTGGGGATTAGGTGGGAGT –3′ and reverse 5′– TCAATCATGTCACTCACTCACTCAA –3′ using the KOD-Plus-Neo DNA polymerase (Novagen) under the following PCR conditions: 3 min at 96˚C followed by 10 s at 98˚C; 30 s at 60 ˚C (Annealing temperatures), 1.5 min at 68 ˚C, and then 10 min at 68 ˚C. This process was repeated for 33 cycles. The first PCR products was used as a template for the second PCR using long primers, *SoFLDH* forward 5′–GGGGACAAG TTTGTACAAAAAAGCAGGCTTCATGTGG GGATTAGGTGGG–3′ and reverse 5′–GGGGACCACTTTG TACAAGAAAGCT GGGTTCAAT

CATGTCACTCACT–3′. BP Clonase (Invitrogen, USA) was used to insert the PCR products into the Gateway entry vector pDONR221. The positive pDONR221- *SoFLDH* constructs harbouring target genes were sequenced and ligated with the destination vector pB2GW7 using Gateway LR Clonase (Invitrogen, USA), then the positive pB2GW7- *SoFLDH* was used for *A. thaliana* plant transformation [3, 4].

### *Arabidopsis* plant growth conditions and preparation of *Agrobacterium* cultures for floral-dip transformation

The ecotype of *A. thaliana* seeds Columbia-0 (Col-0) has prepared for germination by adding 1.2 ml sterilized -water for seeds at a 2.0 ml Eppendorf tube, then nursed at ~ 4˚C for three days at the refrigerator. Then *A. thaliana* seeds had been growing in a growth chamber with humidity of 60–70% under the day and night temperature of 22˚C day/20˚C night, using a light density of 100–150 mol $m^{-2}$ $s^{-1}$ using fluorescent bulbs and photoperiod at 16 h day/8 h night. For floral-dip transformation, plants at two-month age were used, and one week after, the primary inflorescences were clipped. The Plant watering was stopped at four days before the transformation to increase and improve the transformation efficiency. In addition, the constructs of pB2GW7-*SoFLDH* were introduced into *Agrobacterium tumefaciens* strain GV101 by direct electroporation. Recombinant GV101 was grown for 48 h at 28˚C in solid LB media supplemented with 60 μg/ml of each rifampicin and spectinomycin. An individual colony was injected into 0.8 ml of liquid medium and grown at 28˚C under 180 rpm agitation overnight with the same media composition. After 24 h, 0.8 ml of each sample of liquid medium was relocated to a 300 ml conical flask containing 60 ml of LB media supplemented with the same compositions; the samples were grown at 28˚C in a shaker overnight until an optical density of 0.75 (OD$_{600}$) was reached. Overnight cell cultures were harvested by centrifugation at 4,000 rpm for 12 min at 4˚C, and the pellet was resuspended in the floral-dip inoculation medium contained 5.2% sucrose and 0.055% Silwet. *A. thaliana* was transformed by drenched the secondary inflorescences in the inoculation medium and stirred softly to allow the intake of *Agrobacterium* harbouring the pB2GW7-*SoFLDH* vector into the flower gynoecium. The transformed plants were kept in the dark and covered by plastic cover overnight to maintain humidity. After 24 h, the plants were returned back to their normal growth conditions. The transformation was repeated after 7–10 days to increase the transformation efficiency. Plants were grown for additional 30–37 days, until all of the siliques became brown and dry. The seeds were harvested and stored at ~4˚C under desiccation [3, 4, 29, 30]. BASTA was used for selection of transformant seedlings which were also confirmed with PCR for positive transgenic lines, more than 12 positive plant lines with selective gene were analysed for terpenoid profiling and target gene expression.

### Phenotypic evaluation

Transgenic plants were watered and fertilized regularly with Miracle Gro fertilizer (Scott's Company, USA) prepared according to manufacturer's instructions for phenotypic comparisons between *A. thaliana* transgenic lines and their counterpart wild-type plants. Plants were grown in the growth chamber under the previously reported conditions for vegetative growth and flowering. Plants were assessed with regard to leaf morphology, flowering time, and terpene metabolic [4].

### Metabolite extraction from transgenic *A. thaliana* leaves

Terpenoid compounds from overexpression of *SoFLDH- A.thaliana* and wild type were extracted and isolated. Thirty six leaves from transgenic *A. thaliana* line (three leaf from each

plant) were grind in liquid nitrogen with a mortar and pestle to fine powder, and directly soaked in n-hexane as a solvent in Amber storage bottles, 30 ml screw-top vials with silicone/ PTFE septum lids (http://www.sigmaaldrich.com) were applied to diminish the loss of volatiles to the headspace then incubated with shaking at 37˚C and 200 rpm for 72 h. Afterward, the solvent was transmitted using a glass pipette to a 10 ml glass centrifuge tube with screw-top vials with silicone/PTFE septum lids and centrifuged at 5,000 rpm for 10 min at 4˚C to remove plant debris. The supernatant was pipette into glass vials with a screw cap and oil was concentrated until remaining 1.5 ml of concentrated oils under a stream of nitrogen gas with a nitrogen evaporator (Organomation) and water bath at room temperature (Toption-China-WD-12). The concentrated oils transferred to a fresh crimp vial amber glass, 1.5 ml screw-top vials with silicone/PTFE septum lids were used to diminish the loss of volatiles to the headspace. For absolute oil recovery, the remaining film crude oil in the internal surface of concentrated glass vials was dissolved in the minimum volume of n-hexane, thoroughly mixed and transferred to the same fresh crimp vial amber glass, 1.5 ml. And the crimp vial was placed on the auto sampler of the gas chromatography mass spectrometer (GC-MS) system for GC-MS analysis, or each tube was covered with parafilm after closed with screw-top vials with silicone/ PTFE septum lids and stored at -20˚C until GC-MS analysis [3, 4, 31].

## GC-MS analysis of essential oil components

GC analysis was implemented using a Shimadzu model GCMS-QP2010 Ultra (Tokyo, Japan) system. 1μl aliquot of each sample was introduced (split ratios of 15:1) into a GC-MS equipped with an HP-5 fused silica capillary column (30 m x 0.25 mm ID, 0.25 μm film thicknesses). Helium was used as the carrier gas at a constant flow of 1.0 ml/min$^{-1}$. The mass spectra were monitored between 50–450 m/z. Temperature was initially under isothermal conditions at 60˚C for 10 minutes. Temperature was then increased at a rate of 4˚C/min$^{-1}$ to 220˚C, held isothermal at 220˚C for 10 minutes, increased by 1˚C/ min$^{-1}$ to 240˚C, held isothermal at 240˚C for 2 min, and finally held isothermal for 10 minutes at 350˚C. The identification of the volatile constituents was determined by parallel comparison of their recorded mass spectra with the data stored in the Wiley GC/MS Library (10$^{th}$ Edition) (Wiley, New York, NY, USA), the Volatile Organic Compounds (VOC) Analysis S/W software, and the NIST Library (2014 edition). The relative percent amount of each component was calculated by comparing its average peak area to the total areas. All of the experiments were performed simultaneously three times under the same conditions for each isolation technique with total GC running time was 80 minutes [3, 4, 31].

## Results and discussion

### *In silico* analysis of *SoFLDH*

The *SoFLDH* gene with 2,580 bp of open reading frame, which encodes a 860 amino acid and a 95.114 kDa of molecular mass and a 8.59 PI of theoretical isoelectric point (pI). The surmised amino acid sequence of *SoFLDH* showed signal peptide longer than monoterpene synthases (600–650 aa), and other sesquiterpene synthases of 550–580 aa. Furthermore, the presence of 30 amino acid long targets sequence using 'iPSORT' program suggested that SoFLDH protein was localized into the Mitochondrial and chloroplast where sesquiterpene and triterpene biosynthesis takes place. BLASTX analysis revealed that the *Salvia splendens* farnesol dehydrogenase was the nearest homologue gene to *SoFLDH*, with 91.74% identity, 91% Query cover and 0.0 E-value (https://blast.ncbi.nlm.nih.gov/Blast.cgi). Also, from the phylogenetic tree analysis we found the surmised amino acid sequence of *SoFLDH* similar to farnesol dehydrogenase (TEY48599.1) gene from *Salvia splendens*, and other terpene alcohol dehydrogenases and

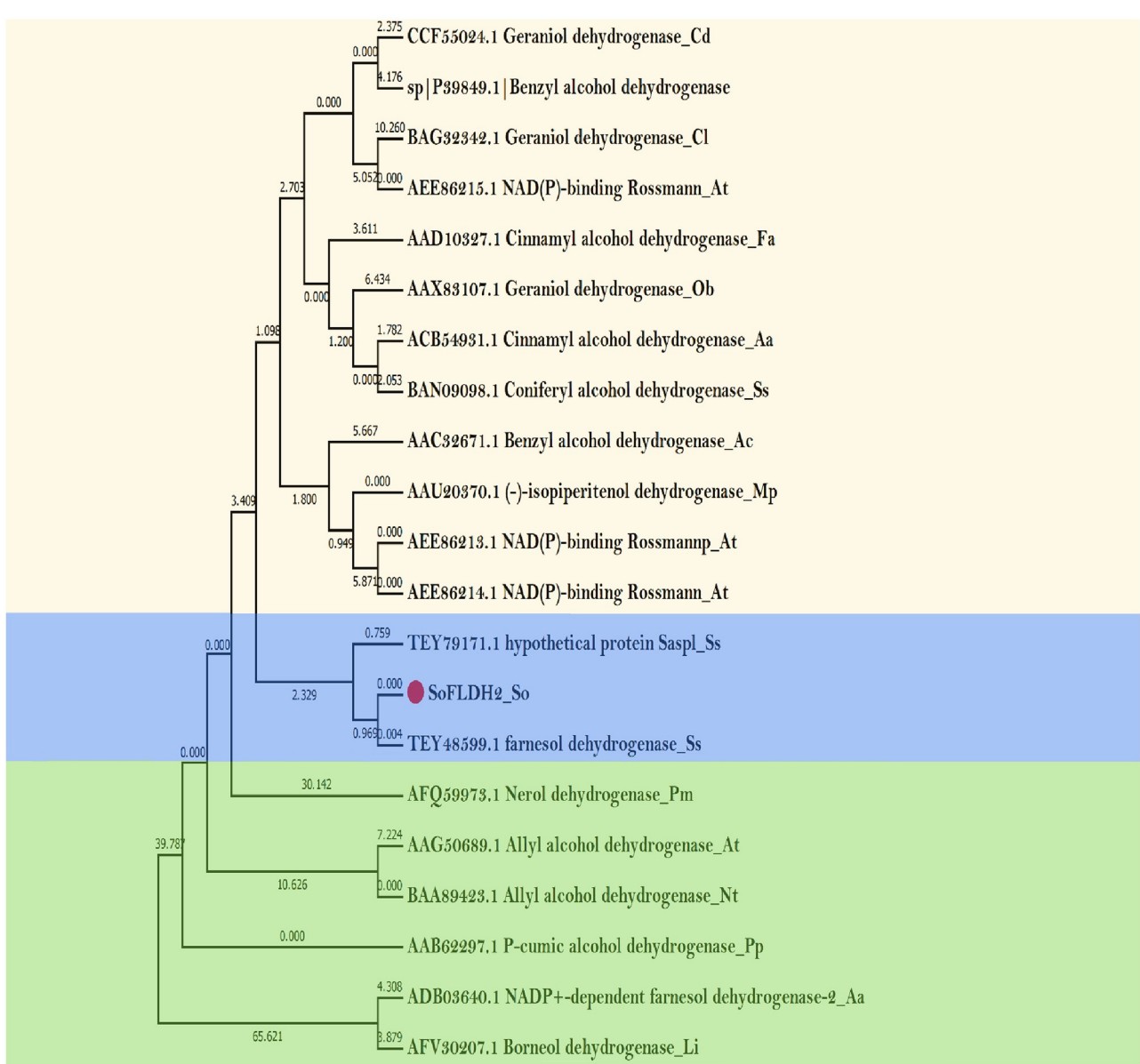

**Fig 1. Phylogenetic analysis of *SoFLDH* gene from *S. officinalis* and other plants, based on the primary amino acid sequences.** A maximum likelihood tree was constructed for *SoFLDH* gene using MEGA 6.6 program with default parameters. The percentage of bootstrap values is provided in the branches. Sequences are labeled with the accession numbers and species names.

benzyl alcohol dehydrogenases genes from different plant species (Fig 1). The function of *SoFLDH* gene was initially foretell depended on sequence alignment with well-known others terpene alcohol dehydrogenases and benzyl alcohol dehydrogenases genes sequences and other conserved motifs from Lamiaceae family and other plants (Fig 2). For instance, SoFLDH protein contained two highly conserved motifs YxxxK (residues 9–13) and (residues 585–589) motifs. Moreover, SoFLDH protein contained other commonly conserved RXR motif (residues 285–287), which is important for product cyclization in Class III TPS proteins [32–34] (Fig 2). Furthermore, SoFLDH protein contained another conserved motif such as, RR (X8) W (residues 405–415) region, which are located in most of the sesquiterpene synthases were

```
                                    YxxxK
AEE86213.1    ------------------------------------------------------------   0
SoFLDH        MWGLGGSHYWGRKESGKVEGIVVVFAWMSSQEKHLKNYVDMYSSRGWNSIVCQPQFLNLF  60
TEY48599.1    MWGLGGSHYWGRKESGKVEGIVVVFAWMSCQEKHLKNYVDMYSSRGWNSIVCQPQFLNLF  60
TEY79171.1    MWGLGGSHYWGRKERGKVEGIVVVFAWMSSQEKHLKNYVDMYSSRGWNSIVCQAQFLNLF  60

AEE86213.1    ------------------------------------------------------------   0
SoFLDH        FPDKAASLAQEIVNELIQENLKGIDLDLSENELKIRPCPIIFASFSGGPKACMYKVLQII  120
TEY48599.1    FPDKAASLAQEIVNELIQENLKGIDLDLSENELKIRPCPIIFASFSGGPKACMYKVLQII  120
TEY79171.1    FPDKAASLAQEIVNELIQENLKGIE-------LKIRPCPIIFASFSGGPKACMYKVLQII  113

AEE86213.1    ------------------------------------------------------------   0
SoFLDH        EGKWEEQINQDECRLVRDSISGYIFDSCPVDFVSDMGNRFFHHQTGLTISRPPLIASWIT  180
TEY48599.1    EGKWEEQINQDECRLVRDSISGYIFDSCPVDFVSDMGNRFFHHQTGLTISRPPLIASWIT  180
TEY79171.1    EGKCEEQINLDECRLVRDSISGYIFDSCPVDFVSDTGNRFFLHQTGLTISRPPLIASWIT  173

AEE86213.1    ------------------------------------------------------------   0
SoFLDH        SGISSTLDTLFLSRFESQRAEYWQTLYSTVSFRAPYLILCSEDDELAPFQIIFNFATRLK  240
TEY48599.1    SGISSTLDTLFLSRFESQRAEYWQTLYSTVSFRAPYLILCSEDDELAPFQIIFNFATRLK  240
TEY79171.1    SGISSTLDALFLSRFESQRAEYWQTLYSTVSFRAPYLILCSEDDELAPFQIIFNFATRLK  233

                                                      RxR
AEE86213.1    ------------------------------------------------------------   0
SoFLDH        NLGADVKLVKWNKSSHVGHFRHHPEEYSASVTELLTKASITYSQRIRQLEGEKMGMEGGH  300
TEY48599.1    NLGADVKLVKWNKSSHVGHFRHHPEEYSASVTELLTKASITYSQRIRQLEGEKMGMEGGH  300
TEY79171.1    NLGADVKLVKWNKSSHVGHFRHHPEEYSAAVTELLSKAAITYSQRIRQLEGEKMGMEGGH  293

AEE86213.1    ------------------------------------------------------------   0
SoFLDH        DEISYPFNGLRKTATMSRDSLHRVNLDLNDYFHVPSSVEYHEDRAVGSIPDESKGRYIPL  360
TEY48599.1    DEISYPFNGLRKTATMSRDSLHRVNLDLNDYFHVPSSVEYHEDRAVGSIPDESKGRYIPL  360
TEY79171.1    DEISYPFNGLRKAAAMSRDSLHRVNLDLNDYFHVPSSVEYHEDRDVGSIPDESKGRYIPL  353
                                             RR (X)8    W
AEE86213.1    ------------------------------------------------------------   0
SoFLDH        SSPPKISAHGVLGEFLFDACVPKNVEDWDLRFSPSMRSAAFASGRRYLNLNPFFWLSAEA  419
TEY48599.1    SSPPKISAHGVLGEFLFDACVPKNVEDWDLRFSPSMRSAAFASGRRYLNLNPFFSLSGSL  420
TEY79171.1    SSPPKISAHGVLGQFLFDACVPKNVEDWDLRFSPSMRSSAFASGRRSSPFNPIKWHSGQE  413
                              TGxxGhaG
AEE86213.1    ----MGPKMPNTETENMKILVTGSTGYLGARLCHVLLRRGHSVRALVRRTSDLSDLPPE-   55
SoFLDH        CL-KVPITLPLTSRRRMLFCVTGASGYLGGRLCHALLDQGYSVKAFVRKSSDVSSLPPPS  478
TEY48599.1    PLES-SNHTATNQPARKVALVTGASGYLGGRLCRALLHQGYSVKAFVRKTSDISSLPPP-  478
TEY79171.1    YEEM-PDYREFSTRKYLDSLVTGASGYLGGRLCHALLHQGYSVKAFVRKTSDVSSLLPPS  472
                  .              ***::****.***:.** :*:**:*:**::**:*.* *
AEE86213.1    ------VELAYGDVTDYRSLTDACSGCDIVFHAAALVEPWLPDPSRFISVNVGGLKNVLE  109
SoFLDH        GDGGGSLQLVYGDVTDYPSLLEAFSGCHVVFHTAALVEPWLPDPSRFSSVNVGGLRNVLK  538
TEY48599.1    SAAGGSLQLVYGDVTDYPSLLEAFSGCHVVFHTAALVEPWLPDPSRFTTVNVGGLRNVLK  538
TEY79171.1    DAADGSLQLVYGDVTDYPSLLEAFSGCHFVFHTAALVEPWLPDPARFTSVNVGGLRNVLK  532
                 ::*.*******.:* :* ***..***:***********:** :******:***:
                                                      YxxxK
AEE86213.1    AVKET----KTVQKIIYTSSFFALGSTDGSVANENQVHNERFFCTEYERSKAVADKMALN  165
SoFLDH        AYKETEMETETIEKIIYTSSFFALGSTDGYIADETQVHPAKHFCTEYEKSKAISDKIALD  598
TEY48599.1    AYKET----ETIEKIIYTSSFFALGSTDGYIADETQVHPAKHFCTEYEKSKAVSDKIALD  594
TEY79171.1    AYTET----ETIEKIIYTSSFFALGSTDGYIADETQVHPAKHFCTEYEKSKALSDKIALD  588
                * .**    :*::****************** :*:*.***  :.******:***::**:**:
```

**Fig 2. Multiple sequence alignment. The deduced amino acid sequence of *SoFLDH* was aligned with homologues identified from the BLASTX analysis.** The conserved motifs YxxxK, RXR, RR(X8) W and TGxxGhaG aremarked. *SoFLDH*: NAD+-dependent farnesol dehydrogenase [*Salvia officinalis*]; farnesol dehydrogenase [*Salvia splendens*: TEY48599.1]; hypothetical protein Saspl_009804 [*Salvia splendens*: TEY79171.1]; NAD(P)-binding Rossmann-fold superfamily protein [*Arabidopsis thaliana*, AEE86213.1].

marked in (Fig 2). Also, SoFLDH protein contained another conserved motif TGxxGhaG (residues 440–447), and all these previous motifs have previously stated to flank the entrance of the active site. The nucleotide-binding motif TGxxxGhG is the highly conserved motif for coenzyme binding and the stabilizing the central β-sheet. Moreover, the YxxxK motif displays the catalytic center [35] and remains responsible for the predilection for NADP(H) over NAD (H) [36, 37]. Finally, each protein sequences have one or two or all of these conserved domains are belong to the terpene synthase family [3, 4, 11, 38–40].

## Putative tissue expression pattern and subcellular localizations of *SgCINS* gene

We analyzed the putative *SoFLDH* gene expression profile maps based on Arabidopsis transcript expression for further understanding the functions of *SoFLDH* gene at different Arabidopsis tissues (Fig 3a–3c). It is clear from the Arabidopsis Electronic Fluorescent Pictograph Expression Profile Browser that the *SoFLDH* gene was expresses at all Arabidopsis tissues. And *SoFLDH* gene was highly expressed at Root then Petals, Sepals and Hypocotyl (Fig 3a). In context, *GmTPS21*, *SoHUMS*, *SoLINS2*, *SoNEOD*, *SgTPSV*, *SgFARD* and *SgGERIS* genes from *G. max*, *S. officinalis* and *S. guaranitica* were reported with higher expression levels in roots and seeds by liu et al., and Ali et al., [3, 4, 41], respectively. Moreover, the putative tissue specific stem epidermis for *SoFLDH* was analyzed and we found the highly expressed was record in top of stem epidermis more than the bottom of stem epidermis (Fig 3b). Furthermore, the putative subcellular localizations of *SoFLDH* was analyzed based on Arabidopsis protein localization for identified the *SoFLDH* synthesis sites at different cell organs (cell plate, cytoskeleton, cytosol, extracellular, golgi, endoplasmic reticulum, plasma membrane, mitochondrion, nucleus, peroxisome, plastid, unclear, unknown and vacuole) (Fig 3c). It is clear from the Arabidopsis Cell Electronic Fluorescent Pictograph subcellular localizations profiles that the *SoFLDH* gene was highly expressed and presented in plasma membrane then endoplasmic reticulum, vacuole, cytosol, mitochondrion, golgi pody and plastid (Fig 3c). These results are in line with Ali et al.,Taniguchi et al., Chen et al., and Wang et al., [26, 42–44] who reported that most of TPSs genes were targeted to the plastid or other cell organelles such as mitochondrion and nucleus.

## Tissue-specific expression of *SoFLDH* gene by quantitative RT-PCR

To determinate the organ-specific expression pattern of *SoFLDH*, we quantified the expression levels of *SoFLDH* transcripts in *S. officinalis* young leaves, old leaves, stems, bud flowers, flowers and roots tissues using qPCR-PCR (Fig 4). From our results, we found the *SoFLDH* gene is expressed in all tissue with distinct expression patterns. In old leaves, *SoFLDH* transcripts gene showed the highest expression levels, followed by stems, flowers, bud flowers, young leaves and roots (Fig 4). Similar results were obtained by Ali et al., [4] of which, the highest expression for sesquiterpene gene encoded by Selinene synthase (*SgTPS-3*) was reported in old leaves, followed by bud flowers [4].

## The 3D structure of SoFLDH protein

SoFLDH protein sequence contains large, conserved domain, which was identified using the InterPro protein sequence analysis & classification (https://www.ebi.ac.uk/interpro/) database. In context that, The SoFLDH protein with a 860-aa length has an NAD(P)-binding domain superfamily domain (IPR036291) from 433–625 aa, and this previous domain have overlapping entries with other domains superfamily, such as Oxidoreductase, N-terminal (IPR000683), Semialdehyde dehydrogenase, NAD-binding (IPR000534), Lactate/malate

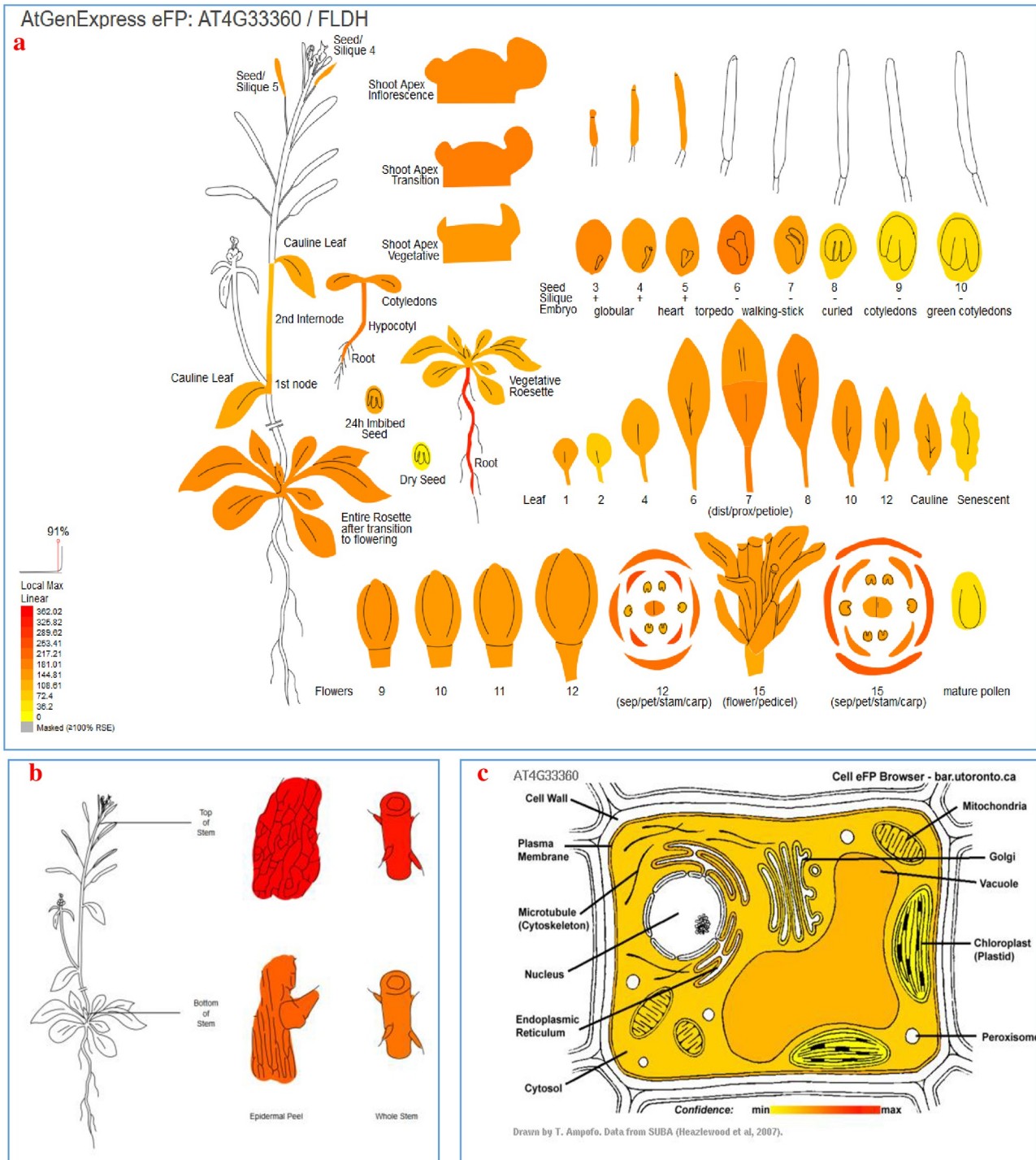

**Fig 3. Visualization the putative an "electronic fluorescent pictograph" browsers for exploring the putative tissue expression and cell localization of *SoFLDH* (AT4G33360) gene, based on Arabidopsis gene expression and protein localization at different tissues and cell organs. a** Expression data at different tissues from seedling to flowering stages. **b** Expression data of tissue specific stem epidermis at top and bottom. **c** Expression data at different cell organs. The blue arrow points the expression scale (the more intense red color, the more gene expression).

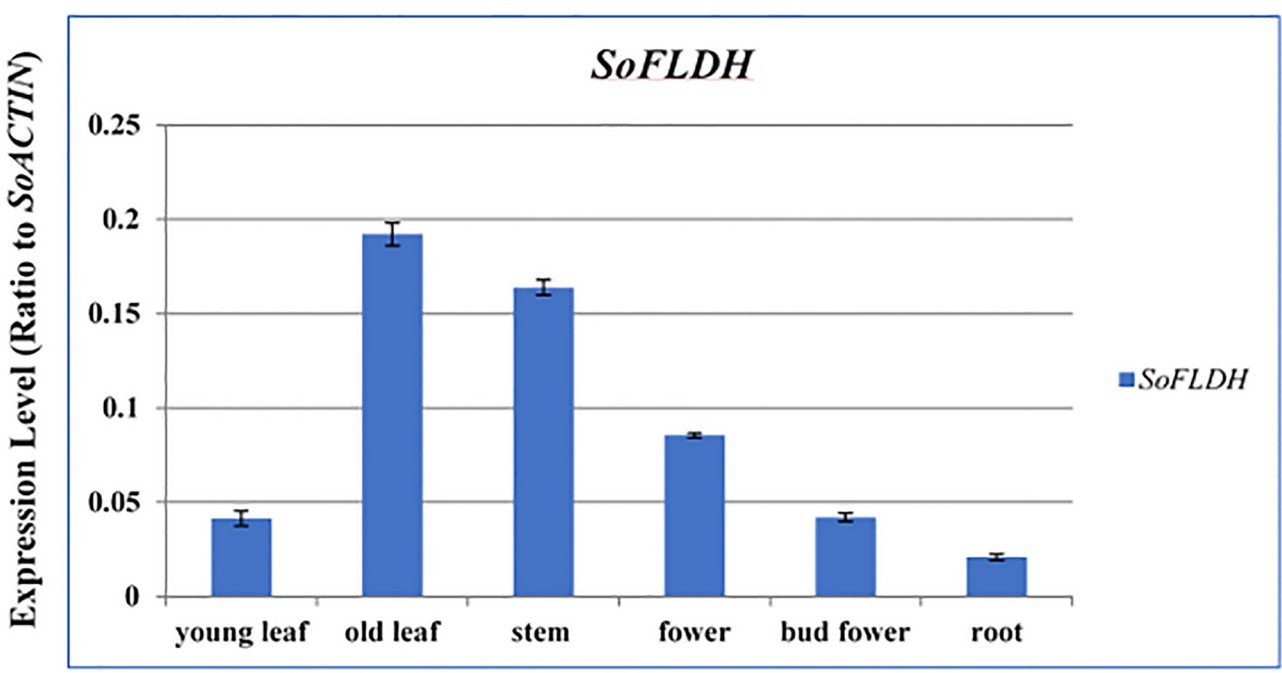

**Fig 4. Quantitative RT-PCR validation of expression of *SoFLDH* gene from various tissues of *S. offcinalis*.** Total RNAs were extracted from young leaves, old leaves, stem, flower, bud fower and root samples and the expression of *SoFLDH* gene was analyzed using quantitative real-time. *SoACTIN* was used as the internal reference. The values are means ±SE of three biological replicates.

dehydrogenase, N-terminal (IPR001236), Dihydrodipicolinate reductase, N-terminal (IPR000846), and NAD-dependent epimerase/dehydratase (IPR001509). This domain is shown in the 3D model built using SWISS-MODELserver (https://swissmodel.expasy.org). The 3D protein model was constructed using UDP-N-acetylglucosamine C4-epimerase (PelX) gene that belong to short-chain dehydrogenase/reductase (SDR) superfamily from *Pseudomonas aeruginosa* [PDB accession: 6wjb] as a template [45] (Fig 5). From the 3D model, SoFLDH was shown to consist entirely of alpha-helices and beta-sheets, long and short connecting loops and turns. Four conserved domains NAD 1: Nicotinamide-Adenine-Dinucleotide, NAD 3: Nicotinamide-Adenine-Dinucleotide, UD1.2: Uridine-Diphosphate-N-Acetylglucosamine and UD1.4: Uridine-Diphosphate-N-Acetylglucosamine were found in the ligands pocket, which encompass the active site (Fig 5).

### Functional characterization of *SoFLDH* gene in transgenic *A. thaliana* leaves

The function and specificity of *SoFLDH* have been detected by *A. thaliana* Columbia-0 (Col-0) transgenic plants. Overexpression of *SoFLDH* in *A. thaliana* was achieved using *Agrobacterium tumefaciens* strain GV101 harboring the transformation vector pB2GW7-*SoFLDH*. Twelve BASTA-resistant transgenic *A. thaliana* were generated with longer flowering stems (Fig 6a and 6b). In contrast, the transgenic *A. thaliana* showed longer flowering stems with many flowers compare with wild-type (Fig 6a). Expression of the *SoFLDH* gene in positive transgenic *A. thaliana* was confirmed using semi-quantitative RT-PCR(Fig 6b). The transcription level of the transgenic plants were verified using Quantitative RT-PCR (Fig 6c). Leaves of twelve 35-day-old transgenic plants and wild types were sampled for RNA isolation and cDNA synthesis. The transgenic plants represent high expression of the *SoFLDH* gene than wild-type.

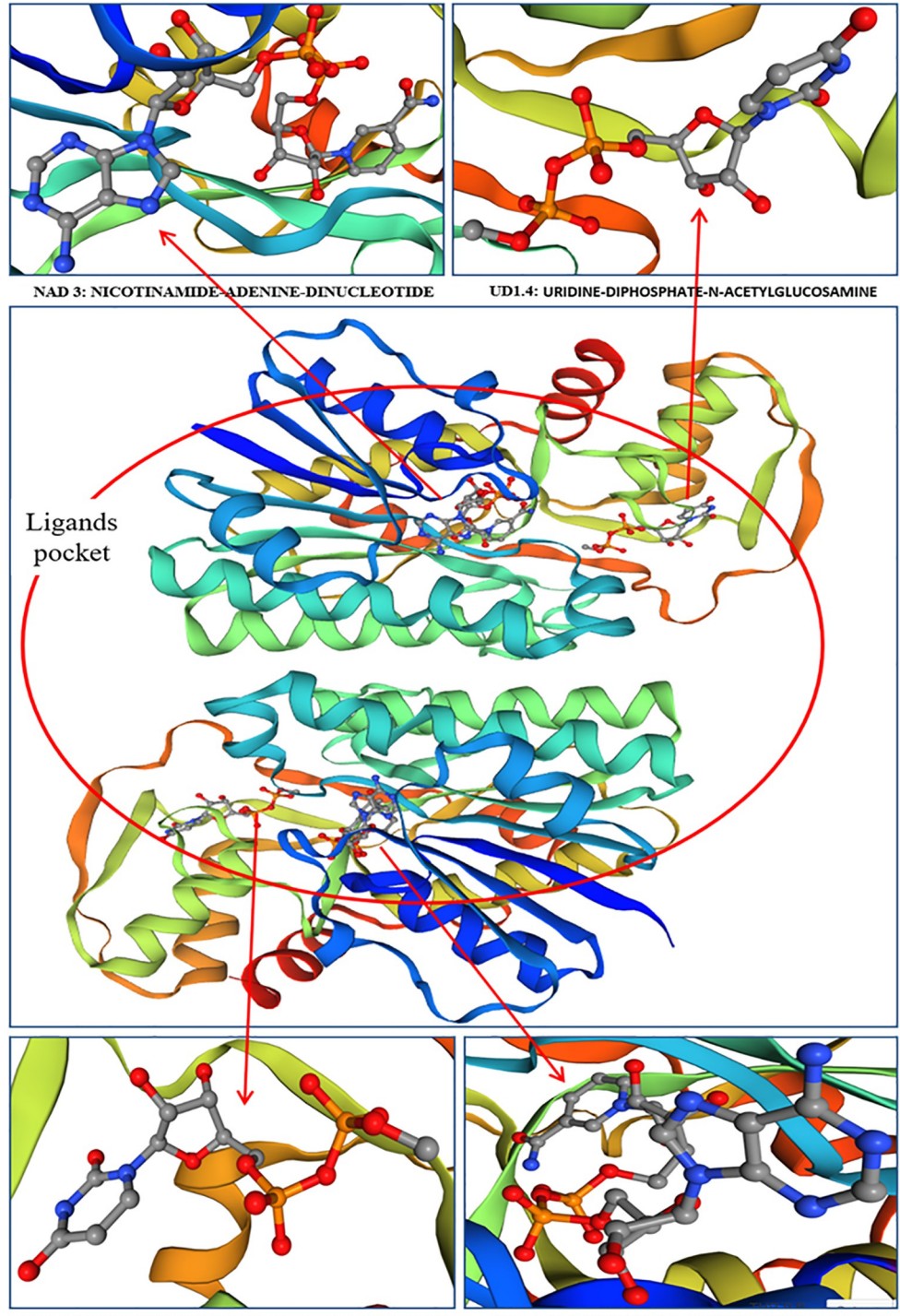

**Fig 5. Predicted 3D model of *SoFLDH* generated by the SWISS-MODEL software.** The arrows represent the ligands pocket with predicted active binding residues, the Nicotinamide-adenine-dinucleotide (NAD) domain and the Uridine-diphosphate-n-acetylglucosamine (UD) domain.

A

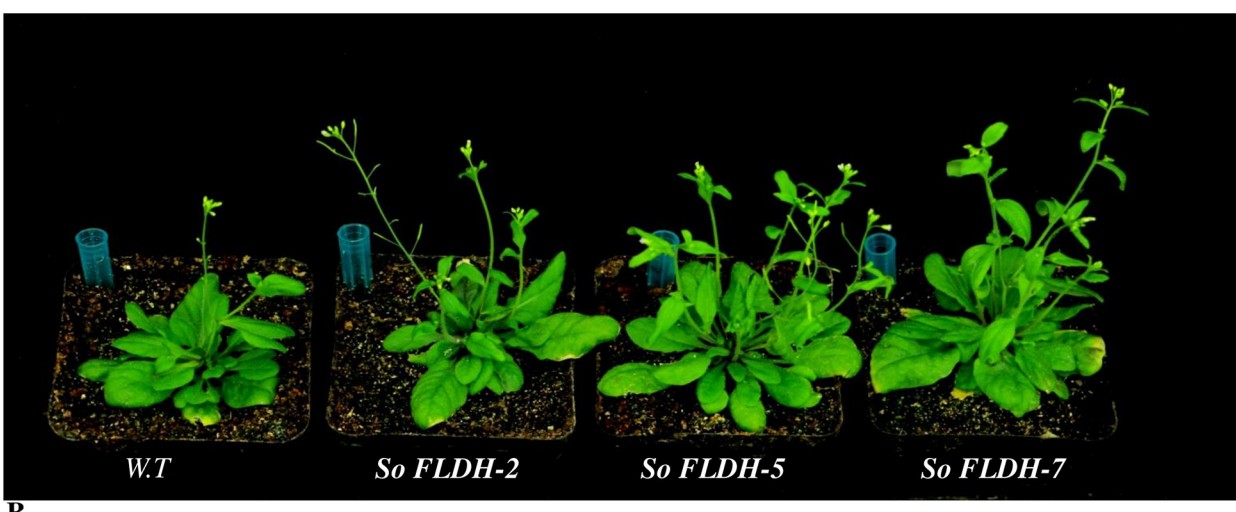

B

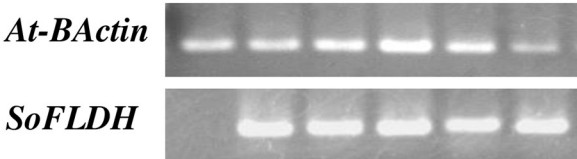

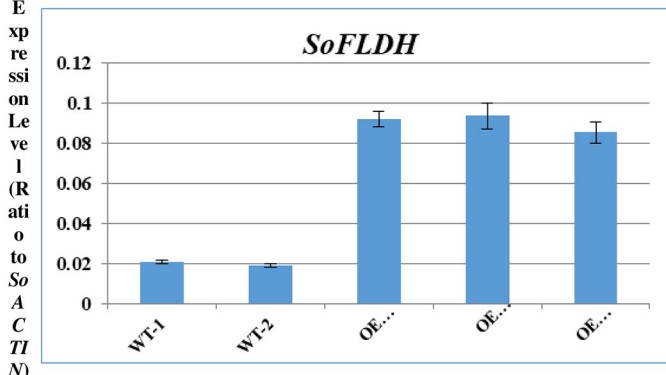

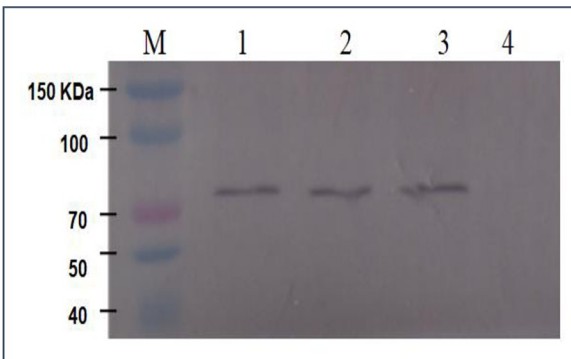

**Fig 6. Overexpression of *SoFLDH* gene in transgenic Arabidopsis.** (A) Comparison of the phenotypes of the transgenic and wild type *A. thaliana*. (B) Semiquantitative RT-PCR to confirm the expression of terpenoid genes. (C) Quantitative RT-PCR validation of expression of *SoFLDH* gene from wild-type and transgenic *A. thaliana* lines. (D) Western blotting (WB) detection of *SoFLDH* protein in transgenic and wild-type *A. thaliana*, the first left-hand most lane represent molecular marker proteins (M), *OE-SoFLDH-2* (lane 1), *OE-SoFLDH-5* (lane 2), *OE-SoFLDH-7* (lane 3) and wild-type (W.T; lane 4).

Moreover, we used Western blotting (WB) analysis for confirmed the transformation stability by *SoFLDH* gene, and protein expression success (Fig 6d). This finding confirmed that the *SoFLDH* gene was overexpressed successfully in *A. thaliana*. Based on the transcription level and WB analysis results, we select *OE-SoFLDH-2*, *OE-SoFLDH-5* and *OE-SoFLDH-7* for further analysis. Meanwhile the morphological analysis resulted in delayed in flowering formation in wild type compared to the transgenic plants (Fig 6a). In context, this results are in line with Ali et al., [3, 4], which found that the overexpression of terpenoids and TPS synthesis genes,

such as *SoLINS, SoNEOD, SoTPS6,, SoSABS, SoCINS, SgGPS, SgFPPS and SgLINS* from *S. officinalis* and *Salvia guaranitica* in *Nicotiana tabacum* and *A. thaliana*, occurred delayed in growth and flowering formation in wild type plants compared to the transgenic plants.

### Terpene contents in transgenic *A. thaliana* leaves

The metabolite was analyzed by GC-MS to recognized the unique terpenes that formed by transformation with the *SoFLDH* gene (S1 Fig). The mono-, sesqui- and diterpene peaks were obviously measured. The type and number of metabolites were displayed by the percentage of peak area (% peak area) (Table 1). We are using the mass spectra libraries, reported references and the extracts of wild-type Arabidopsis which produce disparate amounts and types from terpenoids for identified the terpenes in wild type and transgenic *A. thaliana*. As expected, the leaves of transgenic A. thaliana plants emitted a high level of various terpenoids compare with leaves of wild-type. Furthermore, in transgenic A. thaliana leaves the diterpene compounds were reported as the main group after (30.49%), followed by sesquiterpenes group (15.46%) then one monoterpene compound (0.89%). While, in wild-type A. thaliana the diterpene compounds group was detected as the only group (13.12%) (S1 Fig and Table 1). Moreover, the results shown in S1 Fig and Table 1, represent clear alteration in the transgenic plants, with many new peaks at 29.297, 30.303, 32.328, 35.301, 39.599, 49.213 and 63.242 of retention time. This peak was identified as Levo-β-Elemene, Cis-Caryophyllene, Gamma.-Muurolene, (-)-β-Bourbonene, Cis-Thujanol, Trans-Phytol and Farnesan, based on the nearest hit from a search of the Wiley GC/MS, NIST Library and VOC Analysis S/W software (S1 Fig and Table 1). The production of various terpene by overexpression of *SoFLDH* gene in *A. thaliana*, was getting previously by [4, 29]. With more direct interest in our results, the *SoFLDH* was responsible for the production of various types of terpene specially sesquiterpene through the same common isoprenoid pathway in sesquiterpene biosynthesis. It is worth noting that our results are in lines with several previously evidence supported that various terpene synthases genes have ability to synthesize a number of metabolite simultaneously, like, carene synthases, (±)-linalool synthases, cineole synthases, myrcene synthase, β-amyrin synthases and terpinolene synthases [10, 11, 46–49].

## Conclusions

The present study highlighted to clone and functionally identify one of the narrowly expressed sesquiterpene synthase (*SoFLDH*), which is responsible for the production of NAD+-dependent farnesol dehydrogenase [EC:1.1.1.354] in *S. officinalis*. Overexpression *SoFLDH* in *A. thaliana* resulted in accelerating the growth and flowering of *OE-SoFLDH -2, OE-SoFLDH-5* and *OE-SoFLDH-7* transgenic lines. These three lines exhibited a high expression of the *SoFLDH* gene, which regulate the production of various terpenes. The various types of terpene especially sesquiterpene formed in these *A. thaliana* transgenic plants reveal the dexterity of *A. thaliana* for synthesizing the same product through the common mevalonate pathway of sesquiterpene biosynthesis. While, *SoFLDH* protein exhibits a strong sequence similarity to farnesol dehydrogenase (TEY48599.1) gene from *Salvia splendens*, in addition to other terpene alcohol dehydrogenases and benzyl alcohol dehydrogenases genes from different plant species. Whereas, SoFLDH protein contained four types from highly conserved motifs YxxxK, RXR, RR (X8) W, TGxxGhaG and four conserved domains in the ligands pocket, and these previous domains have overlapping entries with another important superfamily domain. Overall, these data revealed that the *A. thaliana* plant can strongly use as a successfully transformation system for study the terpene synthase genes in *S. officinalis*.

**Table 1. The major Terpenoid compositions in transgenic *A. thaliana* leave over-expressing of *SoFLDH*.**

| N | Compound name | R.T (min.) | Formula | Molecular Mass (g mol-1) | Terpene Type | % Peak area | |
|---|---|---|---|---|---|---|---|
| | | | | | | *AtW.T* | *SoFLDH* |
| 1 | Dimethylsiloxane pentamer | 19.503 | C10H30O5Si5 | 370.7697 | | | 1.16 |
| 2 | Thiourea, tetramethyl- | 22.326 | C5H12N2S | 132.227 | | 1.41 | - |
| 3 | Dodecamethylcyclohexasiloxane | 26.276 | C12H36O6Si6 | 444.9236 | | - | 2.32 |
| 4 | Levo-β-Elemene | 29.297 | C15H24 | 204.3511 | Sesqui | - | 1.9 |
| 5 | Cis-Caryophyllene | 30.303 | C15H24 | 204.3511 | Sesqui | - | 1.54 |
| 6 | Tetradecamethylcycloheptasiloxane | 31.896 | C14H42O7Si7 | 519.0776 | | - | 4.58 |
| 7 | Gamma.-Muurolene | 32.328 | C15H24 | 204.3511 | Sesqui | - | 9.04 |
| 8 | Topanol O | 33.006 | C15H24O | 220.3505 | | - | 2.02 |
| 9 | (-)-β-Bourbonene | 35.301 | C15H24 | 204.3511 | Sesqui | - | 2.2 |
| 10 | Hexadecamethylcyclooctasiloxane | 36.82 | C16H48O8Si8 | 593.2315 | | - | 2.9 |
| 11 | Allethrin | 38.032 | C19H26O3 | 302.4079 | | - | 1.33 |
| 12 | Pyrethrin I | 38.456 | C21H28O3 | 328.4452 | | - | 2.1 |
| 13 | Cis-Thujanol | 39.599 | C10H18O | 154.2493 | Mono | - | 0.89 |
| 14 | Octadecamethylcyclononasiloxane | 41.04 | C18H54O9Si9 | 667.3855 | | - | 2.23 |
| 15 | Phytan | 41.196 | C20H42 | 282.5475 | Diter | 2.76 | - |
| 16 | Hexahydrofarnesyl acetone | 42.676 | C18H36O | 268.4778 | | - | 1.95 |
| 17 | Oleic Acid | 43.799 | C18H34O2 | 282.468 | | 3.00 | - |
| 18 | Cyclohexasiloxane, dodecamethyl- | 44.792 | C12H36O6Si6 | 444.9236 | | - | 1.31 |
| 19 | Palmitic acid | 45.31 | C16H32O2 | 256.4241 | | 31.16 | - |
| 20 | (Z)-9-Tetradecenal | 46.587 | C14H26O | 210.3556 | | - | 1.41 |
| 21 | Palmitic acid, trimethylsilyl ester | 47.316 | C19H40O2Si | 328.6052 | | 5.69 | - |
| 22 | Methyl dehydroabietate | 47.469 | C21H30O2 | 314.4617 | | - | 2.49 |
| 23 | Heneicosane | 48.625 | C21H44 | 296.5741 | | 3.73 | - |
| 24 | Trans-Phytol | 49.213 | C20H40O | 296.531 | Diter | - | 30.49 |
| 25 | Trans-Elaidic acid | 49.488 | C18H34O2 | 282.4614 | | 29.59 | - |
| 26 | Dodecanoyl chloride | 49.889 | C12H23ClO | 218.763 | | - | 10.35 |
| 27 | Heptadecane, 8-methyl- | 50.911 | C17H36 | 240.4677 | | 1.84 | - |
| 28 | Octadecamethylcyclononasiloxane | 51.454 | C18H54O9Si9 | 667.3855 | | - | 1.12 |
| 29 | Cadinane | 51.916 | C20H41Cl | 316.993 | Diter | 10.36 | - |
| 30 | 4-Methyldodecane | 54.362 | C13H28 | 184.3614 | | - | 1.11 |
| 31 | Stigmasterol acetate | 57.289 | C31H50O2 | 454.7275 | | - | 1.05 |
| 32 | Octadecamethylcyclononasiloxane | 61.974 | C18H54O9Si9 | 667.3855 | | - | 0.77 |
| 33 | Farnesan | 63.242 | C15H32 | 212.4146 | Sesqui | - | 0.78 |
| 34 | Bis(2-ethylhexyl) phthalate | 64.443 | C24H38O4 | 390.5561 | | - | 6.5 |
| 35 | Dihomo-γ-linolenic acid; (Z,Z,Z)-icosatri-8,11,14-enoic acid | 65.99 | C20H34O2 | 306.4828 | | - | 1.37 |
| 36 | Heneicosane | 74.674 | C21H44 | 296.5741 | | - | 3.97 |
| 37 | 2,6,10,14-Hexadecatetraen-1-ol, 3,7,11,15-tetramethyl-, acetate, (E,E,E)- | 77.731 | C22H36O2 | 332.5200 | | - | 1.12 |
| 38 | Tetrapentacontane | 78.973 | C54H110 | 759.4512 | | 10.46 | - |
| | Total % Peak area | | | | | % 100 | % 100 |
| | Total Precentage of Monoterpenes | | | | | - | 0.89 |
| | Total Precentage of Sesquiterpenes | | | | | - | 15.46 |
| | Total Precentage of Diterpenes | | | | | 13.12 | 30.49 |

## Supporting information

**S1 Fig. Typical GC-MS mass spectrographs for terpenoids from leaf of *A. thaliana* plants.** (DOCX)

## Acknowledgments

Our sincere appreciation for supporting of the Academy of Scientific Research and Technology (ASRT) with the National Natural Science Foundation of China and with the Egyptian Deserts Gene Bank, Desert Research Center. The authors thank Professor Dr. Osama Ezzat Elsayed and Dr. Aamir Hamid Khan for proof-reading the manuscript.

## Author Contributions

**Conceptualization:** Mohammed Ali.

**Formal analysis:** Mohamed Hamdy Amar.

**Methodology:** Mohammed Ali, Elsayed Nishawy, Mohamed Ewas, Ahmed H. M. Hassan, Shengwei Wang.

**Project administration:** Guang-Wan Hu.

**Resources:** Mokhtar Said Rizk, Ahmed G. M. Sief-Eldein.

**Supervision:** Mohamed Hamdy Amar, Qing-Feng Wang.

**Validation:** Walaa A. Ramadan, Mokhtar Said Rizk, Mohamed Abd S. El-Zayat, Mingquan Guo, Fatma A. Ahmed.

**Visualization:** Elsayed Nishawy.

**Writing – original draft:** Mohammed Ali, Elsayed Nishawy, Shengwei Wang.

**Writing – review & editing:** Elsayed Nishawy, Mohamed Hamdy Amar, Qing-Feng Wang.

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
