## [Decision Letter · Decision Letter 0]

2 Feb 2022

PONE-D-22-00853Molecular characterization of a Novel NAD+-dependent farnesol dehydrogenase  SoFLDH  gene involved in sesquiterpenoid synthases from Salviaofficinalis.PLOS ONE

Dear Dr. Amar,

Thank you for submitting your manuscript to PLOS ONE. After careful consideration, we feel that it has merit but does not fully meet PLOS ONE’s publication criteria as it currently stands. Therefore, we invite you to submit a revised version of the manuscript that addresses the points raised during the review process.

We look forward to receiving your revised manuscript.

Kind regards,

Jen-Tsung Chen, Ph.D.

Academic Editor

PLOS ONE

Journal Requirements:

"This study was funding by the National Natural Science Foundation of China and with Academy of Scientific Research and Technology (ASRT) under the collaboration project between Wuhan Botanical Garden, Chinese Academy of Sciences, Wuhan, 430074, China, and Desert Research Center (DRC), Cairo, Egypt."

"This work was supported and financial by the Academy of Scientific Research and Technology (ASRT) with the National Natural Science Foundation of China."

"This study was funding by the National Natural Science Foundation of China and with Academy of Scientific Research and Technology (ASRT) under the collaboration project between Wuhan Botanical Garden, Chinese Academy of Sciences, Wuhan, 430074, China, and Desert Research Center (DRC), Cairo, Egypt."

Reviewers' comments:

Reviewer's Responses to Questions

**Comments to the Author**

1. Is the manuscript technically sound, and do the data support the conclusions?

Reviewer #1: Partly

Reviewer #2: Partly

Reviewer #3: Yes

2. Has the statistical analysis been performed appropriately and rigorously? 

Reviewer #1: Yes

Reviewer #2: N/A

Reviewer #3: Yes

3. Have the authors made all data underlying the findings in their manuscript fully available?

Reviewer #1: No

Reviewer #2: Yes

Reviewer #3: Yes

4. Is the manuscript presented in an intelligible fashion and written in standard English?

Reviewer #1: Yes

Reviewer #2: No

Reviewer #3: No

5. Review Comments to the Author

Reviewer #1: 1. Authors needs to reconstruct phylogeny based on Maxmium likelihood analysis. Please see:

The BAHD Gene Family in Cacao (Theobroma cacao, Malvaceae): Genome-Wide Identification and Expression Analysis

Magnesium transporter Gene Family: Genome-Wide Identification and Characterization in Theobroma cacao, Corchorus capsularis and Gossypium hirsutum of Family Malvaceae

2. The authors did not mentioned about the transformation stability. The authors did not used Southern or Northern blotting analysis experiment. The primers used in PCR is also not mentioned. The authors need to justify the choice in the discussion.

I think the data of the manuscript is good but these two point give very much limitation to the study.

Reviewer #2: The present study cloned and functionally verified one of the narrowly expressed sesquiterpene synthase (SoFLDH) genes, responsible for producing NAD+-dependent farnesol dehydrogenase in S. officinalis. Overexpression SoFLDH in A. thaliana resulted in accelerating the growth and flowering of OE-SoFLDH -2, OE-SoFLDH-5, and OE-SoFLDH-7 transgenic lines. These three lines exhibited a high expression of the SoFLDH gene, which regulates the production of various terpenes. Mainly, the methodology, results, and discussion need to be improved.

1. Add abstract in the main text. Also, add the key findings and take-home message in 1-2 sentences at the end of the abstract.

2. Line 48, please specify the growth conditions.

3. Line 53, PROTPARAM is an online tool/website, not software.

4. Line 84-85, also add “following the manufacturer's instructions.

5. One of the major issues is the presentation of the results and discussion. Combining results and discussion can be considered. However, the authors must divide the contents into subheadings for more clarity. Also, the current discussion is quite superficial and must be improved accordingly.

6. The results regarding metabolites can be better explained, not just describe the data. Also, explain what does dataset suggests. Thus, I strongly suggest dividing the results into different subheadings and improving the results and discussion presentation.

7. There are spacing and spelling mistakes. The authors are requested to proofread the whole text.

8. The English quality can further be improved.

Reviewer #3: - ENGLISH of the manuscript needs to polish and be improved.

- Line 63-64: please rewrite this sentence.

- All gene names should be in italic format.

- Line 90: "85°C for 5 second", it is 5 second or 5 min? I think it is 5 min. Please check it.

- Line 100-102: After 33 cycles, the final extension (10 min at 68°C) was applied. Please consider it again.

- Line 223, 229, 238, etc.: protein name should not be italic.

- Line 253: these instated to this.

- Line 253: after "by" provide the names of the authors.

6. PLOS authors have the option to publish the peer review history of their article (what does this mean?). If published, this will include your full peer review and any attached files.

Reviewer #1: No

Reviewer #2: No

Reviewer #3: No

---

## [Author Response · Author response to Decision Letter 0]

15 Apr 2022

Dear editor,

Thank you for your useful comments. We have been revised the manuscript “Molecular characterization of a Novel NAD+-dependent farnesol dehydrogenase SoFLDH gene involved in sesquiterpenoid synthases from Salvia officinalis” and make corrections required by the reviewers and the detailed information are listed as attached file 

Responses to reviewers

---

## [Decision Letter · Decision Letter 1]

13 May 2022

Molecular characterization of a Novel NAD+-dependent farnesol dehydrogenase  SoFLDH  gene involved in sesquiterpenoid synthases from Salviaofficinalis.

PONE-D-22-00853R1

Dear Dr. Amar,

We’re pleased to inform you that your manuscript has been judged scientifically suitable for publication and will be formally accepted for publication once it meets all outstanding technical requirements.

Kind regards,

Jen-Tsung Chen, Ph.D.

Academic Editor

PLOS ONE

Additional Editor Comments (optional):

Reviewers' comments:

Reviewer's Responses to Questions

**Comments to the Author**

1. If the authors have adequately addressed your comments raised in a previous round of review and you feel that this manuscript is now acceptable for publication, you may indicate that here to bypass the “Comments to the Author” section, enter your conflict of interest statement in the “Confidential to Editor” section, and submit your "Accept" recommendation.

Reviewer #1: All comments have been addressed

Reviewer #2: All comments have been addressed

2. Is the manuscript technically sound, and do the data support the conclusions?

Reviewer #1: Yes

Reviewer #2: (No Response)

3. Has the statistical analysis been performed appropriately and rigorously? 

Reviewer #1: Yes

Reviewer #2: (No Response)

4. Have the authors made all data underlying the findings in their manuscript fully available?

Reviewer #1: Yes

Reviewer #2: (No Response)

5. Is the manuscript presented in an intelligible fashion and written in standard English?

Reviewer #1: Yes

Reviewer #2: (No Response)

6. Review Comments to the Author

Reviewer #1: (No Response)

Reviewer #2: The authors have satisfactorily addressed all the comments. Thus, the revised version can be accepted for publication.

7. PLOS authors have the option to publish the peer review history of their article (what does this mean?). If published, this will include your full peer review and any attached files.

Reviewer #1: No

Reviewer #2: No

---

## [Editor Report · Acceptance letter]

24 May 2022

PONE-D-22-00853R1 

Molecular characterization of a Novel NAD^+^-dependent farnesol dehydrogenase *SoFLDH* gene involved in sesquiterpenoid synthases from *Salvia officinalis.*

Dear Dr. Amar:

I'm pleased to inform you that your manuscript has been deemed suitable for publication in PLOS ONE. Congratulations! Your manuscript is now with our production department. 

Kind regards, 

on behalf of

Dr. Jen-Tsung Chen 

Academic Editor

PLOS ONE